# Engaging Private Health Care Providers to Identify Individuals with TB in Nepal

**DOI:** 10.3390/ijerph182211762

**Published:** 2021-11-09

**Authors:** Rajesh Sah, Upendra Kumar Singh, Ranju Mainali, Ataulhaq Sanaie, Tripti Pande, Nathaly Aguilera Vasquez, Amera Khan

**Affiliations:** 1Save the Children International, Bardibas 45701, Nepal; rajesh.sah@savethechildren.org; 2Sahayog Samittee Nepal, Kalaiya 44412, Nepal; ranjumainali90@gmail.com; 3Independent Consultant, London E11 4DP, UK; Sanaie.as@gmail.com; 4McGill Knowledge Management, McGill International TB Center, Montreal, QC H3G 1A4, Canada; tripti.pande@mail.mcgill.ca (T.P.); nathaly.aguileravasquez@mail.mcgill.ca (N.A.V.); 5Stop TB Partnership, TB REACH—Innovations and Grant Team, 1218 Geneva, Switzerland; amerak@stoptb.org

**Keywords:** tuberculosis, private health care providers, public private mix

## Abstract

In Nepal, 47% of individuals who fell ill with TB were not reported to the National TB Program in 2018. Approximately 60% of persons with TB initially seek care in the private sector. From November 2018 to January 2020, we implemented an active case finding intervention in the Parsa and Dhanusha districts targeting private provider facilities. To evaluate the impact of the intervention, we reported on crude intervention results. We further compared case notification during the implementation to baseline and control population (Bara and Siraha) notifications. We screened 203,332 individuals; 11,266 (5.5%) were identified as presumptive for TB and 8077 (71.7%) were tested for TB. Approximately 8% had a TB diagnosis, of whom 383 (56.2%) were bacteriologically confirmed (Bac+). In total, 653 (95.7%) individuals were initiated on treatment at DOTS facilities. For the intervention districts, there was a 17%increase for bacteriologically positive TB and 10% for all forms TB compared to baseline. In comparison, the change in notifications in the control population were 4% for bacteriologically positive, and −2% all forms. Through engagement of private sector facilities, our intervention was able to increase the number of individuals identified with TB by over 10% in the Parsa and Dhanusha districts.

## 1. Introduction

Annually, tuberculosis (TB) affects over 10 million individuals worldwide [1]. Nepal, like its two neighboring countries, India and China, is plagued with a high TB burden. Based on a recent National TB Prevalence Survey in 2018, there are about 117,000 people living with TB in Nepal resulting in a reported prevalence rate of 416 per 100,000 population, 1.8 times higher than previously estimated [2]. In addition to underestimation of TB burden in the country, Nepal also faces challenges with underreporting of TB cases. In 2018, there were 69,000 individuals who fell ill with TB in Nepal, however only 32,474 (47%) were reported to the National TB Program [3]. This may be due to a variety of reasons including extended and costly travel to health facilities, poor TB knowledge and initial care-seeking in the private sector [4].

To address underreporting from the private sector, a key recommendation from Nepal’s 2018 National Prevalence Survey is the establishment of a mandatory TB notification system in the private sector [2]. In Nepal, the private sector finances approximately 65% of health care [5]. Patient pathway analyses have shown that approximately 60% of persons with TB initially seek care in the private sector upon developing symptoms [6]. Previous studies have noted that in urban settings of Nepal, 50% of individuals with TB were poorly managed by the private sector where staff is often not adequately trained in TB management and care [5,7]. Further, a study conducted among private providers in Nepal noted that only 27% of private providers maintained a complete record of the individuals with TB whom they diagnosed and/or treated [5]. These findings underscore the importance of private sector engagement in controlling the TB epidemic in Nepal.

In efforts to improve TB screening for those accessing care in the private sector, Sahayog Samittee Nepal (SS Nepal), a non-profit organization in Nepal aiming to ensure the right to health care for all individuals, designed and implemented a private sector engagement intervention. This intervention, supported by a TB REACH grant, aimed to intensify TB case finding in private healthcare provider facilities, including private physicians and pharmacies. This evaluation presents the crude results of our intervention and overall additional TB notifications following the implementation period.

## 2. Methods

### 2.1. Setting

From November 2018 to January 2020, we developed and implemented a private sector active case finding (ACF) intervention in two districts, Parsa and Dhanusha, henceforth evaluation population (EP). Two other districts, Bara and Siraha, with similar socio-demographics, population size and economic indicators were selected as control districts, henceforth control population (CP). This was done to help evaluate the results of the intervention by comparing CP and EP notification trends (see Figure 1).

Dhanusha and Parsa are two border districts located in Province 2 in southeastern Nepal. Province 2 borders India and has an all forms TB case notification rate (CNR) of 92 per 100,000 population [8]. Despite being the province with the fourth highest CNR in the country, the private sector only contributes to 17% of case finding in the province, which is the second lowest contribution rate, only preceded by Sudurpaschim (12%) [8].

In Dhanusha, the intervention was implemented in two municipalities: Janakpur (also known as Janakpursham) and Sabaila, both located southeast of Kathmandu. Janakpur is a sub-metropolitan city in Dhanusha district and has a population of approximately 173,924, which is nearly 25% of the district’s total population. Sabaila has a population of 24,893, which represents 3% of the district’s total population. In Parsa, the intervention was implemented in Birgunj and Pokhariya, located south of Kathmandu. Birgunj is a metropolitan city with a population of 240,922, representing 35% of the district’s total population. Pokhariya is a municipality with a population of 32,885, which is nearly 5% of the district’s total population.

### 2.2. Intervention

Prior to implementing the intervention, we mapped and engaged private provider facilities in the two Parsa and Dhanusha districts to establish “cough screening desks” (CSDs). Private health facility staff, hereafter health volunteers (HV), were placed at CSDs to screen individuals seeking care for TB (Figure 2).

HVs approached all attendants of private provider facilities for TB screening. Ultimately, only consenting individuals were screened using a paper-based screening questionnaire, which included questions on TB symptoms (cough ≥ 2 weeks, fever, night sweats, loss of appetite, weight loss, and/or presence of blood in sputum), previous history of TB and contact with persons with TB. An individual was identified as presumptive for TB if they had one or more TB-like symptoms, previous history of TB and/or had recently been in contact with a person with confirmed TB. Consenting individuals identified as being presumptive for TB were asked to provide a sputum sample if Xpert testing was available or two sputum samples in case of microscopy testing. One sputum sample was taken on-the-spot, and another was taken one-hour later. If sputum could not be provided by the individual identified as presumptive, they were referred for clinical examination and CXR by a physician. Health mobilizers (HM), recruited by SS Nepal staff, delivered the sputum samples from each CSD to NTP laboratories for sputum smear or Xpert (depending on availability) evaluation using motorbikes. Those individuals confirmed for TB, either clinically or bacteriologically, were referred to public Directly Observed Treatment Short-Course (DOTS) facilities for treatment initiation and TB notification.

Each private provider received a performance-based incentive of 200 Nepalese rupee (NPR), approximately 1.70 USD, per individual confirmed with TB and 10 NPR (0.08 USD) for each sputum sample collected. Each HV received 1.5 NPR per individual screened at the CSD. The laboratory personnel at the NTP laboratories received 10 NPR (0.08 USD) per sputum slide examined. HMs received a monthly salary and allowance for their motorbike fuel.

### 2.3. Data Collection, Analysis, and Evaluation

The methodology for evaluating the intervention followed the established TB REACH monitoring and evaluation framework [9]. To assess the impact of the intervention, the framework compares TB notifications from the EP during the timeframe of the project to; (1) historic notifications in the intervention districts prior to project implementation and to (2) notifications from the CP where the intervention was not implemented.

As part of the evaluation, we established a set of indicators that were collected from the participating health facilities in the intervention districts. Indicators, disaggregated by district, included the number of individuals screened, tested for TB, bacteriologically confirmed, clinically confirmed, initiated on treatment, and successfully treated. Data from each of the participating private health care facilities in the intervention districts were collected using paper-based screening forms that were filled by HV’s at the CSD. Each HV was also responsible for entering all patient information, testing results and referral into a presumptive TB register at the CSD. Every month, HVs tabulated the data from the presumptive TB register to send to the Project Coordinator via mobile phone. During monthly meetings, HVs also brought and submitted the paperversion of the tabulated indicators to the Project Coordinator. The Project Coordinator digitized the aggregate indicators into an Excel sheet and sent them for approval to the District Health Officers (Authorized Staff of NTCC). Aggregate data from all CSDs wereentered and tabulated on Excel 2016. Additional analyses were undertakenusing R Studio.

TB case notifications (bacteriologically confirmed and clinically diagnosed) were collected from the NTP registers for the previous three years for both the EP and CP. To achieve this, SS Nepal was granted credentials to access the NTP District Health Information Software (DHIS) 2 which contains TB notification data for all districts. Changes in TB notifications for both the intervention and control districts were calculated from the difference between the historical and intervention period and the trend was calculated using simple regression analysis. A test of proportions was used to assess whether the additionality (representing the change in case notifications) in the EP was significantly different from that in the CP.

## 3. Results

In total, we mapped 115 private health facilities in the Parsa and Dhanusha districts, of which 63 (55%) were engaged by the intervention. There were 27 physicians, 30 pharmacies/auxiliary health workers (AHWs), and 6 laboratories offering outpatient department (OPD) services. We engaged 63 HVs and 4 HMs. From November 2018 to January 2020, we screened 203,332 individuals for TB, of whom 109,874 (53.5%) were male and 93,458 (46.5%) were female (see Table 1). No refusals for screening were recorded. Among these individuals, 11,266 (5.5%) were identified as presumptive for TB of whom 8077 (71.7%) were tested for TB. The main reasons for not being bacteriologically tested for TB were loss to follow-up and inability to produce sputum. Individuals who received a clinical diagnosis or who had extrapulmonary TB were included in the all forms (AF) category, as well as those who had a diagnosis via sputum smear or Xpert for pulmonary TB. The bacteriologically confirmed (Bac+) category is limited to those who received sputum smear or Xpert diagnosis. Approximately 8% (682) were confirmed for TB, of whom 383 (56.2%) were Bac+. Among those who were confirmed for TB, 431 (63.5%) were male and 251 (36.5%) were female. In total, 653 (95.7%) of individuals were initiated on treatment at DOTS facilities and 540 (82.7%) completed treatment. As shown in Table 1, throughout the intervention more individuals were screened, tested and treated in Dhanusha compared to Parsa. Further, there were more males (431, 63.5%) diagnosed with TB in comparison to females (251, 36.5%).

Overall, the intervention resulted in an 8% increase in the number of individuals identified with TB in Parsa and 13% in Dhanusha compared to baseline (Table 2). Table 2 further comparesthe change in notifications between the baseline period of 16 November 2016 to 15 November 2018 (exactly one year prior to initiation of the intervention) and after implementation of the intervention (16 November 2018 to 15 January 2020). In the EP, there was a 17%increase for Bac+ and10%for all forms AFcompared to baseline. In comparison, the change in notifications in the CP were 4% for Bac+ and −2% AF.Test of proportion results demonstrate that the changes observed for Dhanusha and Parsa on both Bac+ and AF case notifications compared to the observed changes in the CP are significant (*p* < 0.01).

## 4. Discussion

Our project was able to engage with over half of the private sector facilities that we mapped in our intervention districts. Through our engagement of private sector facilities, we were able to increase the number of individuals identified with TB by 10% in Parsa and Dhanusha districts compared to baseline which demonstrates the need for interventions engaging the private sector in Nepal. Our results highlight the fact that TB affects males more than females. While the proportion of males and females screened as well as tested were approximately the same, more males (63.5%) were diagnosed with TB in comparison to females (36.5%). This is reflected in other literature indicating that men represent 57% of the people who develop TB, in comparison to women who represent 32% [10]. This may be due to risk-exposing occupations, care seeking behaviors, or biological differences [10,11,12].

Of the 8077 microbiological tests conducted (i.e., Xpert MTB/RIF or sputum smear microscopy), only 383 (5.0%) were bacteriologically confirmed. In a study conducted by Nepal et al., 32 (6.8%) individuals were confirmed as bacteriologically positive for TB among 468 microbiological tests conducted (5). This suggests that our bacteriological positivity rate was lower than expected. This may indicate poor quality of sputum sample production and/or examination. To improve sputum quality, additional education on how to produce effective sputum samples for individuals with presumptive TB should be provided [13].

Further, we found that while Bac+ notifications remained slightly lower in the EP (1472 versus 1524) post-implementation, AF notifications were higher in the EP (2670) as compared to those in the CP (2494). Although there could be various reasons for this, it is possible that providing the opportunity for CXR screening to individuals who were symptomatic but could not provide a sputum sample contributed to all forms TB detection. According to Nepal’s recent TB prevalence survey, 70% individuals identified with TB did not have TB-like symptoms and were only identified by chest X-ray (CXRs) [3]. Although the current intervention did not focus on CXR referral, SS Nepal scaled up their private sector engagement intervention in January 2020 and has placed a bigger emphasis on CXRs. SS Nepal is currently collecting data on CXR referrals and outcomes.

Further, our intervention results showcase a higher level of individuals screened, tested, diagnosed, and subsequently treated in Dhanusha district in comparison to Parsa district. We found that more private providers agreed to participate in the intervention in Dhanusha, thus more CSDs were implemented which led to more individuals screened. Further, there was a higher proportion of Bac+ found among tested in Dhanusha (66.3%) than in Parsa (33.7%). One of the reasons for this could be that in Parsa there is only one GeneXpert machine, however more are present in Dhanusha, thus more individuals were tested using GeneXpert, while in Parsa more individuals were tested with sputum smear. Specifically, at the time of the implementation, there were four Xpert machines in Dhanusha: one at Janakpur District Health Office laboratory, one at Yadukoha Primary Health Care Centre (PHC), one at PHC of Sabaila Municipality and one at Dhalkebar Health Post. In Parsa, the only Xpert machine was located at Narayani central hospital.This underscores the importance of increasing accessibility to GeneXpert machines in Nepal to increase TB case detection.

### 4.1. Lessons Learnt

As this intervention was implemented as a proof-of-concept, the accumulated lessons learnt are important to highlight. First, certain providers were situated far away from NTP laboratories, rendering sputum transport by HM much more difficult. In these cases, the project team recommended that individuals identified as presumptive at those locations be referred to nearby NTP laboratories for sputum collection by NTP staff. At the NTP laboratories, there were instances of supply chain issues causing shortage of reagents for microscopy testing and/or lack of Xpert cartridges. These issues were addressed through careful coordination and collaboration with the NTP as well as ensuring communication with other NTP laboratories that could be used as backup. Alternative NTP laboratories were also used when GeneXpert machines were out of service. This highlights the importance of ensuring robust laboratory networks, as well as strong engagement of local NTP staff to enable quick action when such challenges arise. Further, there was unexpected staff turnover among the HVs, which was addressed through continuous re-orientation and training of staff. There was also some reluctance from individuals with TB symptoms to provide sputum since it was not prescribed by their physician. Such concerns were appeased through education and counselling from the HVs. Lastly, difficulties in ensuring treatment enrollment for individuals who did not have access to a phone or lived outside the intervention districts were resolved through close communication with the DOTS center staff.

### 4.2. Limitations

While our evaluation has highlighted the strengths of our intervention, there were some limitations. First, we did not document the experiences of the private providers who engaged with our intervention. Our intervention showcases the successes of engaging private providers; however, to enable successful planning of future interventions, we require knowledge on the experiences of private providers to ensure their requests are integrated into future approaches. Secondly, our intervention only engaged with private provider facilities/clinics and did not engage with public sector clinics. For this reason, the total number of individuals identified with TB in these two districts may have been higher if screening had been carried out in public facilities as well. Nevertheless, we aimed to engage private providers due to previous findings indicating a high prevalence of initial care seeking in the private sector [6]. Further, we only engaged with two municipalities within each district. Since the TB REACH grant received was aimed at proof-of-concept of the approach employed, only a limited number of districts were involved. However, given the success of this intervention in increasing TB case finding in Dhanusha and Parsa’s private sectors, SS Nepal has received a second TB REACH grant to scale up the intervention to three other districts. Another limitation is that the project was not able to account for individuals who were visiting the CSDs who lived outside the EP, thus certain individuals may have been found presumptive in the CSDs in the EP but notified as cases in the CP, which may have diluted the yield of the intervention. It was also not possible to distinguish notifications from the public and private sector for the intervention, thus we could only report on case notifications integrating both sectors. Future studies should consider disaggregating notifications by public and private sector to enable evaluation of intervention on private sector notifications. Additionally, the CP had higher notifications than the EP despite similar population and sociodemographic characteristics. We believe that this could be due to ongoing interventions from another organization providing TB services in many districts including Bara and Siraha (CP), where there was ongoing Global Fund support to increase ACF in government facilities. At the time of implementation, there was an ongoing intervention, IMPACT TB, which also aimed to increase TB case detection in four districts in Nepal, including Dhanusha [14]. This intervention, implemented by the Birat Nepal Medical Trust, could also account for part of the increase in case notifications in this district. It is important to note that this could partially explain the higher increase in case notifications seen in Dhanusha compared to Parsa (13% versus 8%). This could have resulted in a synergistic effect of both interventions implemented in the same period, thus the increase in notifications cannot be solely associated with the SS Nepal intervention.

## 5. Conclusions

Our evaluation showcases the impact of engaging private providers in TB screening and diagnosis. Through the presence of CSDs directly at the private provider clinic, we were able to screen a significant proportion of individuals. Private providers are often the first point of contact for many individuals seeking care and integrating TB screening into their facilities was proven to increase TB case detection and notification in two urban districts of Nepal. Our intervention demonstrated a 10% increase in TB case notifications. To further increase the level of involvement of private providers, qualitative studies understanding their experiences with active case finding interventions are required. These will enable implementors to provide holistic interventions, which are not only beneficial to the individuals with TB, but also facilitate engagement with private providers. Further, similar successful interventions should be piloted and evaluated in the country, specifically in rural areas of Nepal where populations have limited access to health services. Proof-of-concept interventions such as this one also present important opportunities to compile lessons learnt and to share with the TB community to provide recommendations to improve and strengthen future ACF implementation.

## Figures and Tables

**Figure 1 ijerph-18-11762-f001:**
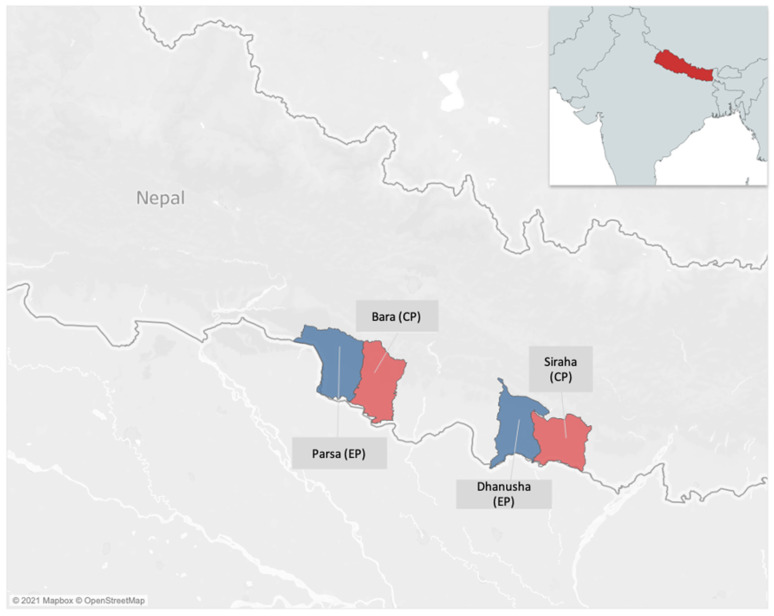
Evaluation and control populations for SS Nepal’s TB REACH intervention.

**Figure 2 ijerph-18-11762-f002:**
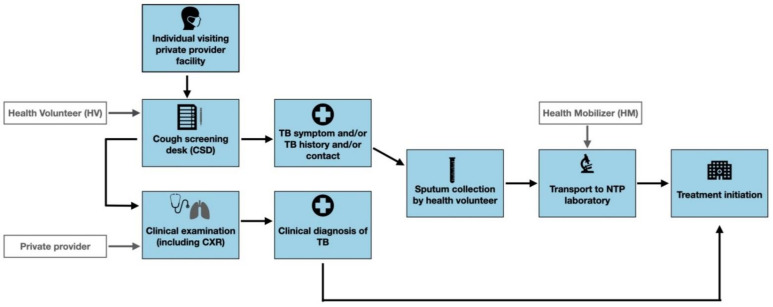
SS Nepal intervention patient pathway.

**Table 1 ijerph-18-11762-t001:** Process indicators disaggregated by gender and location (November2018 to January2020).

Process Indicators	Parsa		Dhanusha		Total
	*n*	%	*n*	%	
Number of people screened	49,766	24.5%	15,3566	75.5%	203,332
Number of people with presumptive TB	2591	23.0%	8675	77.0%	11,266
Number of people tested	1858	23.0%	6219	77.0%	8077
Number of people with Bac+ TB ^1^	129	33.7%	254	66.3%	383
Number of people diagnosed with AF ^2^ TB	249	36.5%	433	63.5%	682
Number of individuals with Bac+ TB started on treatment	123	33.6%	243	66.4%	366
Number of Individuals with Bac+ TB completedTreatment	107	34.6%	202	65.3.1%	309
Number of individuals with AF TB started on treatment	243	37.2%	410	62.8%	653
Number of Individuals with AF TB started on Treatment	209	38.7%	331	61.2%	540

^1^ Bac+ = bacteriologically confirmed TB cases; ^2^ AF = all forms of TB (bacteriologically confirmed, clinically confirmed and extra-pulmonary TB).

**Table 2 ijerph-18-11762-t002:** Unadjusted additionality in EP (Dhanusha and Parsa) and CP (Bara and Siraha).

Population	Baseline—Case Notification		Implementation—Case Notification		Additionality		% Change from Baseline		*p*-Value for Bac+	*p*-Value for AF
	Bac+ ^a^	AF ^b^	Bac+ ^a^	AF ^b^	Bac+ ^a^	AF ^b^	Bac+ ^a^	AF ^b^		
EP ^1^	1262	2422	1472	2670	210	248	17%	10%		
*Dhanusha*	610	1109	767	1257	157	148	26%	13%	0.04	<0.01
*Parsa*	652	1313	705	1414	53	101	8%	8%	0.04	<0.01
CP ^2^	1465	2548	1524	2494	58	−54	4%	−2%		

^1^ EP = evaluation population (Parsa and Dhanusha); ^2^ CP = control population (Bara and Siraha); ^a^ Bac+ = bacteriologically confirmed TB cases; ^b^ AF = all forms of TB (bacteriologically confirmed, clinically confirmed and extra-pulmonary TB); Test of proportions, significance defined at *p* < 0.05.

## Data Availability

Not applicable.

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
