# Peer review of "Engaging Private Health Care Providers to Identify Individuals with TB in Nepal"

_ijerph, 2021, doi:10.3390/ijerph182211762_

Round 1

Reviewer 1 Report

Overall an important and well presented study on a PPM intervention in Nepal. However some clarifications would help with interpretation of the data presented. 

Please give population of Parsa/Dhanusha and control districts.

-Give the location of 2 municipalities covered by the intervention- what proportion of the district population do these municipalities represent?

-The authors state there is 1 xpert machine in Parsa and 'more in Dhanusha'. Give the precise number and location relative to the intevention sites. 

-Please add disaggreagated data by district.

-There was an active case finding intervention, IMPACT TB, active in Dhanusha district during the same timeframe (hence the extra Xpert machines in Dhanusha district). Please decsribe the TB interventions active in each district and disaggragate data to allow a clear comparison.

-the authors state two sputum collected from each patinet- were two samples tested by Xpert, and if so, what was the rate of discrpepant results?

-Please give number of intervention completing treatment, as well as no. starting treatment.

-please elaborate on interventions active in Bara and Siraha mentioned by the authors. 

Author Response

Dear reviewer,

We thank you for all your comments and taking the time to review our paper. Your feedback has been essential in improving our work. Below, you will find responses to your comments.

  1. Please give population of Parsa/Dhanusha and control districts.

This has been added between lines 71 to 79 on page 2.

  1. Give the location of 2 municipalities covered by the intervention- what proportion of the district population do these municipalities represent?

Both districts are located in Province 2 in southeastern Nepal (line 65, page 2). Specific municipality names, proportion of district population and location are stated in lines 71 to 79, page 2.

  1. The authors state there is 1 Xpert machine in Parsa and 'more in Dhanusha'. Give the precise number and location relative to the intervention sites. 

This information has been added to lines 241 to 244, page 7.

  1. Please add disaggregated data by district.

Data was previously disaggregated in Tables 1 and 2. We are not sure whether this is what is being referred to by disaggregate data by district? Is it possible to provide clarification on this point? We would be happy to make the appropriate changes.

  1. There was an active case finding intervention, IMPACT TB, active in Dhanusha district during the same timeframe (hence the extra Xpert machines in Dhanusha district). Please describe the TB interventions active in each district and disaggregate data to allow a clear comparison.

Thank you for indicating this intervention to us. It has been discussed, along with its possible repercussion on case notification data presented, on line 296 to 304, page 8.

  1. the authors state two sputum collected from each patinet- were two samples tested by Xpert, and if so, what was the rate of discrpepant results?

Thank you for bringing this up. One sputum sample was requested if Xpert testing was available while 2 samples were requested if microscopy was done. This was clarified on lines 109-110, page 4.

  1. Please give number of intervention completing treatment, as well as no. starting treatment.

Thank you for this comment. Treatment completion numbers have been added to Table 1.

  1. please elaborate on interventions active in Bara and Siraha mentioned by the authors. 

Thank you for this comment. In these districts (like in many districts in the country), there was on-going Global Fund support to increase case finding. In Bara and Siraha, this was done in government facilities. This has been further clarified on lines 296-297, page 8.

Reviewer 2 Report

The manuscript reports the results of an intervention study to sensitize for TB case notifications two private sector facilities. The effect was measured against two control district with no intervention.  All steps of the cascade of care were additionally measured. The research was funded under the TB REACH programme. The intervention was associated with a 17% increase in Smear positive TB cases, and 10% of other forms of TB. The manuscript is generally well written and straight forward in the discussion of the results. The result section, and in particular the tables should be substantially modified.

Specific comments

In the method section please specify whether all attendants to the study health care centers were screened for TB of just subgroups.

In the result section the authors should report the numbers of all the steps of the cascade of care in the two control areas.

Table 1: This table is difficult to read and to interpret. Why putting in the same table gender differences and location differences? This is confusing. Moreover, why presenting separately the results of the two intervention studies? I would suggest to show in this table a comparison between before and after intervention, comparing intervention sites and control sites. In addition, statistical tests should be inserted to evaluate the difference among the two settings. Finally, the table needs formatting

 Table 2: this table is probably useless if Table 1 is modified according to the suggestion. Once again, I would suggest to combine the results of the two intervention sites, as are combined the results of the two control sites

Figure 3: there might something wrong in the figure. Visually, the reader cannot appreciate any difference in TB notifications before and after the intervention and between intervention and control areas. The legend of the “y” ax should be added

Author Response

Dear reviewer,

We thank you for all your comments and taking the time to review our paper. Your feedback has been essential in improving our work. Below, you will find responses to your comments.

  1. In the method section please specify whether all attendants to the study health care centers were screened for TB of just subgroups.

Thank you for this feedback. All attendants were approached for screening, however only those who consented were taken through the screening questionnaire. Those who did not consent were not screened. No specific sub-group was targeted. This has been clarified on lines 102 to 103, page 4.

  1. In the result section the authors should report the numbers of all the steps of the cascade of care in the two control areas.

Thank you for this comment. However, since the intervention was not implemented in the control areas, we are not able to provide cascade of care results for Bara and Siraha as we do not have access to this data. We do report in Table 2 on the NTP case notification data in both districts.

  1. Table 1: This table is difficult to read and to interpret. Why putting in the same table gender differences and location differences? This is confusing. Moreover, why presenting separately the results of the two intervention studies? I would suggest to show in this table a comparison between before and after intervention, comparing intervention sites and control sites. In addition, statistical tests should be inserted to evaluate the difference among the two settings. Finally, the table needs formatting

Thank you for this feedback. The gender and study sites are presented in the same table to avoid repetition given that separating both would entail presenting on the same process indicators. We would prefer to maintain the table, but would be comfortable creating 2 separate tables if the reviewer believes this would be best. We are presenting on the study sites because we believe that it illustrates some interesting differences which are discussed later in the paper. Given that we do not have cascade of care data for the control districts since an intervention was not implemented in those districts which were primarily selected to provide a basis for comparison of case notification in districts which did not have access to the intervention, it is not possible to make this comparison in Table 1. We have provided a comparison of case notification data for both evaluation and control populations in Table 2 which is presented separately as Table 1 presented results from the intervention itself, but Table 2 is presenting case notification data from NTP official reporting systems. This allows us to compare case notification for a baseline period for both control and evaluation populations to the implementation period and observe whether there is a larger increase in the evaluation population where the intervention was implemented. Regarding statistical tests, given that this paper is looking to evaluate the overall implementation results of this intervention and was not set up as a controlled study powered for statistical tests, we are concerned that conclusions from such test would be inaccurate. Further, we aimed to follow the official TB REACH reporting framework which has been used for many similar interventions in the past. We hope that this addressed this concern, we would be happy to discuss further and provider clarification on these aspects.

  1. Table 2: this table is probably useless if Table 1 is modified according to the suggestion. Once again, I would suggest to combine the results of the two intervention sites, as are combined the results of the two control sites

We believe it is best to maintain both tables separate since Table 1 is relating information on the intervention results while Table 2 is presenting district case notification data.

  1. Figure 3: there might something wrong in the figure. Visually, the reader cannot appreciate any difference in TB notifications before and after the intervention and between intervention and control areas. The legend of the “y” ax should be added

Thank you for this comment. We have added a label to the Y axis. As noted on lines 193 to 196 (page 6), before the intervention, case notification in the control population was higher than in the evaluation population which we also discuss on lines 293 to 297 (page 8). This graph is meant to complement Table 2 and illustrate that after the intervention, despite some fluctuation, there is a slight increase in case notification in the evaluation population.

Round 2

Reviewer 2 Report

Previous question: In the method section please specify whether all attendants to the study health care centers were screened for TB of just subgroups (page 4, line 96).

Thanks for specifying that 1) all attendants were approached for screening and only those who consented were taken through the screening questionnaire, and 2) that no specific sub-group was targeted. Please add the proportion of attendants refusing screening. If this is not known, please state it.

Previous question on Table 1: This table is difficult to read and to interpret. Why putting in the same table gender differences and location differences? This is confusing. In addition, statistical tests should be inserted to evaluate the difference among the two settings. Finally, the table needs formatting

You clarified that the gender and study sites are presented in the same table to avoid repetition. However, I suggest to delete information on gender, which is not essential.

Regarding statistical tests, you replied that given that this paper is looking to evaluate the overall implementation results of this intervention and was not set up as a controlled study powered for statistical tests, we are concerned that conclusions from such test would be inaccurate. I understand that the study was not design to have a specific power, however a statistical test to appraise the differences in rate would still be useful

Previous question on Figure 3: there might something wrong in the figure. Visually, the reader cannot appreciate any difference in TB notifications before and after the intervention and between intervention and control areas.

You replied that, as noted on lines 193 to 196 (page 6), before the intervention, case notification in the control population was higher than in the evaluation population which we also discuss on lines 293 to 297 (page 8). This graph is meant to complement Table 2 and illustrate that after the intervention, despite some fluctuation, there is a slight increase in case notification in the evaluation population. I still believe that the table does not reflect any change in notifications. Could you try to make a figure using percent change from baseline rather that number of TB cases notified?

Please explain in footnote of tables and figures all abbreviations: i.e. B+ and AF in Table 2.

Author Response

Dear Reviewer,

We thank you for your thoughtful comments which have helped improve our manuscript. Below you will find point-by-point responses to each comment.

  1. COMMENT: Previous question: In the method section please specify whether all attendants to the study health care centers were screened for TB of just subgroups (page 4, line 96). Thanks for specifying that 1) all attendants were approached for screening and only those who consented were taken through the screening questionnaire, and 2) that no specific sub-group was targeted. Please add the proportion of attendants refusing screening. If this is not known, please state it.

Thank you for your comment. Throughout the intervention, we did not record any refusals for screening. We have indicated this on line 174.

  1. COMMENT: Previous question on Table 1: This table is difficult to read and to interpret. Why putting in the same table gender differences and location differences? This is confusing. In addition, statistical tests should be inserted to evaluate the difference among the two settings. Finally, the table needs formatting. You clarified that the gender and study sites are presented in the same table to avoid repetition. However, I suggest to delete information on gender, which is not essential.

Thank you for this helpful comment. We do agree that having data disaggregated by both gender and district was confusing. We have removed the gender data from Table 1. Regarding the statistical tests, we did run a test of proportions on the additionality for which we have data for both the CP and EP. This is because process indicator data it not available for the CP, thus a comparison was not possible for Table 1. Lastly, regarding the formatting, this is the journal’s template, is there a specific aspect of the formatting that should be addressed?

  1. Regarding statistical tests, you replied that given that this paper is looking to evaluate the overall implementation results of this intervention and was not set up as a controlled study powered for statistical tests, we are concerned that conclusions from such test would be inaccurate. I understand that the study was not design to have a specific power, however a statistical test to appraise the differences in rate would still be useful

Thank you for this comment. We ran a test of proportions to evaluate whether additionality was significantly different in the EP and CP. We have described this in the Methods section on lines 152-154. Results are outlines on lines 199-211 and in Table 2.

Previous question on Figure 3: there might something wrong in the figure. Visually, the reader cannot appreciate any difference in TB notifications before and after the intervention and between intervention and control areas.You replied that, as noted on lines 193 to 196 (page 6), before the intervention, case notification in the control population was higher than in the evaluation population which we also discuss on lines 293 to 297 (page 8). This graph is meant to complement Table 2 and illustrate that after the intervention, despite some fluctuation, there is a slight increase in case notification in 

  1. the evaluation population. I still believe that the table does not reflect any change in notifications. Could you try to make a figure using percent change from baseline rather that number of TB cases notified?

Thank you for bringing this up. When discussing your comments, we realized that this may not be the best representation of the results as it is confusing, and Table 2 reflects the % change best. Thus, we have decided to remove Figure 3.

  1. Please explain in footnote of tables and figures all abbreviations: i.e. B+ and AF in Table 2.

This has been added to the footnotes of Table 2.

Kind regards,

Upendra Kumar Singh.
